# Treatment of Polycyclic Aromatic Hydrocarbons in Oil Sands Process-Affected Water with a Surface Flow Treatment Wetland

**Alexander M. Cancelli** and **Frank A. P. C. Gobas** *

The School of Resource and Environmental Management, Simon Fraser University, 8888 University Drive, Burnaby, BC V5A 1S6, Canada; alexander_cancelli@sfu.ca

* Correspondence: gobas@sfu.ca

**Abstract:** This study applied a passive sampling approach using low-density polyethylene passive samplers to determine the treatment efficiency of the Kearl surface flow treatment wetland for polycyclic aromatic hydrocarbons (PAHs) in Oil Sands Process-affected Waters (OSPW). Treatment efficiency was measured as concentration-reduction and mass-removal from the OSPW. The results show that the wetland's ability to remove individual PAHs from the influent varied substantially among the PAHs investigated. Treatment efficiencies of individual PAHs ranged between essentially 0% for certain methylated PAHs (e.g., 2,6-dimethylnaphthalene) to 95% for fluoranthene. Treatment in the Kearl wetland reduced the combined total mass of all detected PAHs by 54 to 83%. This corresponded to a reduction in the concentration of total PAHs in OSPW of 56 to 82% with inflow concentrations of total PAHs ranging from 7.5 to 19.4 ng/L. The concentration of pyrene in water fell below water quality targets in the Muskeg River Interim Management Framework as a result of wetland treatment. The application of the passive samplers for toxicity assessment showed that in this study PAHs in both the influent and effluent were not expected to cause acute toxicity. Passive sampling appeared to be a useful and cost-effective method for monitoring contaminants and for determining the treatment efficiency of contaminants in the treatment wetland.

**Keywords:** treatment wetlands; oil sands process-affected water; polycyclic aromatic hydrocarbons; passive sampling; toxicity

## 1. Introduction

As demand for freshwater conservation grows, there is a need for sustainable solutions to manage and reuse process-affected waters. In Canada, a considerable volume of Oil Sand Process-affected Waters (OSPW) has been and continues to be generated during bitumen extraction, which contains an array of different organic and inorganic contaminants [1]. Polycyclic aromatic Hydrocarbons (PAHs) are commonly associated with OSPW and have been shown to be a potential source of OSPW toxicity [2–4].

Since OSPW is currently subject to a 'zero discharge' policy and few treatment options are available, OSPW is either recycled for further use in the extraction process or stored in effluent tailings ponds. These effluent tailings ponds are susceptible to leaching and erosion, and present adverse risks to migratory birds and wildlife that confuse these areas for safe ecological havens [5–8]. While efforts to develop feasible solutions for OSPW treatment are ongoing, few have been realized to date. Treatment wetlands have emerged as a potentially feasible option to treat OSPW [9–15]. Treatment wetlands are constructed, artificial ecosystems that harness the biogeochemistry of natural systems to reclaim and remediate contaminated land and water.

The biogeochemical mechanisms for contaminant removal within a wetland include microbial and plant-mediated biotransformation, chemical transformations, UV degradation, evapotranspiration, and sorption to sediments [12]. The capability of wetlands to harness these biogeochemical processes for wastewater treatment has been demonstrated for a variety of wastewaters including municipal and domestic wastewaters, agricultural runoff, pulp and paper wastewater, and waters that contain surfactants, solvents, or pesticides (e.g., [11,16–22]). While treatment has been demonstrated for a variety of wastewaters, many studies have reported treatment wetland performance based on general metrics for water quality such as Biochemical Oxygen Demand (BOD$_5$), Total Nitrogen, Total Phosphorus, or Total Petroleum Hydrocarbons. Although useful as measures of overall water quality, these general metrics do not detail the removal of specific contaminants of concern from wastewaters. Quantifying the removal of individual contaminants by wetland treatment is critical to evaluate the toxicological risk associated with the influent OSPW and effluent water and to identify which contaminants are more easily removed from wetlands and which contaminants are not, i.e., high vs. low treatment efficiency. Information on the treatment efficiency of engineered treatment wetlands is necessary for assessing the feasibility of treatment wetlands for their specific wastewater challenges.

The objective of this study is to investigate the capacity of wetlands to treat PAHs in OSPW. Specifically, we investigate the treatment efficiency of PAHs in terms of reductions in concentrations, mass loadings, and associated toxicity of PAHs in OSPW in the Kearl Treatment (KT) wetland.

## 2. Materials and Methods

### 2.1. Site Description

The KT wetland is a free water surface-flow constructed wetland at the Kearl Oil Sands site (approximately 75 km NNE of Fort McMurray, AB, Canada; 57°26′00″ N, 111° 8′31″ W) managed by Imperial Oil Resources Ltd. The KT wetland operates in warmer summer months, typically from May to September. It was designed as a pilot-scale wetland to investigate the treatment of on-site Oil Sands Process-affected Water (OSPW). Water is pumped into the wetland at 5 L/s (430 m$^3$/day), resulting in a hydraulic retention time of approximately 14 days.

OSPW refers to any water that comes in contact with oil sands or used in an oil sands processing facility. In 2017, OSPW was collected as runoff from an overburden disposal area. This overburden disposal area contains stockpiles of excavated overburden from the mined areas at the Kearl Oil Sands site and contains residual amounts of organic contaminants. Since this OSPW is not sourced directly from a tailings pond, these residual contaminants largely consist of PAHs. Concentrations of naphthenic acids and dissolved solids in this OSPW were low, hence the focus on PAHs in this study. The runoff is first directed to, and detained in, the north overburden disposal pond located next to the KT wetland which acts as an initial settling basin for suspended solids. Due to the 'zero-discharge' policy for all OSPW, the system operates as a closed-circuit and therefore water is recycled from the KT wetland back into the north overburden disposal pond. The outflow was controlled by a submerged pump that was triggered on when water depth reached 1.7 m, and off when water depth receded to 1.0 m in the final deep pool (Figure 1). In 2018, OSPW for the KT wetland was sourced from a drainage pond situated next to an effluent tailings area at the Kearl Oil Sands. The OSPW was pumped to the wetland during a single pump event, where it was fully recycled within the wetland (i.e., no external detention pond was used) for the duration of the 2018 study.

The wetland consists of six cells in series (3 deep pools, 3 shallow areas) with a longitudinal slope of 0.014%. Water percolates over shallow interior berms that separate adjacent cells. Shallow berms parallel to water flow in the middle of the shallow pool sections were constructed to improve directional water flow, and provide access for wetland monitoring (e.g., vegetation, erosion, water quality). The KT wetland operates at a total volume of water of approximately 6000 m$^3$. The deep pools (forebay, deep pool 1, and final deep pool) operate at a depth of 1.7 m, and are dominated by submerged vegetation, but contain a band of emergent vegetation (approximately 5% of area) along the perimeter of the cells

where water depth is shallower. The shallow area cells are densely vegetated with a variety of different species dominated by common cattails (*Typha latifolia*) and water sedge (*Carex aquatilis*).

The rooting medium consists of 200 mm of compacted peat soil underneath 300 mm of non-compacted peat soil from the displaced muskeg originally found on location. The peat was placed over a geosynthetic clay liner that was blanketed with a non-woven geotextile (Figure 1).

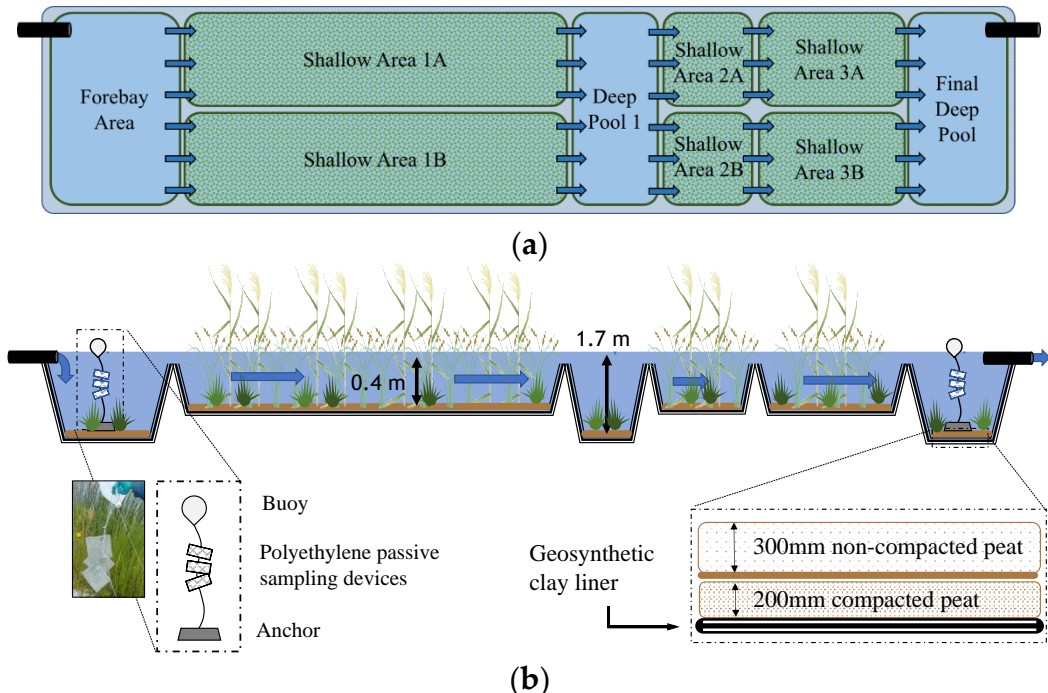

**Figure 1.** Schematic diagram of the Kearl Treatment Wetland showing (**a**) planar view, and (**b**) cross-sectional view with the passive sampling devices and rooting medium. Photo shows polyethylene passive samplers in aluminium mesh casings connected to buoys.

## 2.2. Water Quality Monitoring

Biochemical Oxygen Demand ($BOD_5$), conductivity, Dissolved Inorganic Carbon (DIC), Dissolved Oxygen (DO), Dissolved Organic Carbon (DOC), pH, Total Dissolved Solids (TDS), Total Suspended Solids (TSS), turbidity, and water temperature ($T_{water}$) were collected for the OSPW in the wetland to evaluate changes in the quality of OSPW as it passes through the KT wetland. In the 2017 study, samples were obtained from the forebay and final deep pool on 13 July and 17 August 2017. In the 2018 campaign, OSPW was fully recycled through the wetland, therefore samples were obtained from the forebay on both 26 August and 19 September 2018. Water samples were analysed by Maxxam Analytics (Calgary, AB, Canada) for $BOD_5$, Dissolved Inorganic Carbon (DIC), Dissolved Organic Carbon (DOC), Total Dissolved Solids (TDS), Total Suspended Solids (TSS). Field measurements were collected by WorleyParsons Ltd. (Calgary, AB, Canada) for conductivity, Dissolved Oxygen (DO), pH, and water temperature using a YSI® Professional Plus Multiparameter instrument, and turbidity was measured using an Orbeco-Hellige® TB200™ Turbidimeter.

## 2.3. Passive Sampling

Low-density polyethylene (PE) passive samplers were deployed in triplicate (*n* = 3) in the forebay and final deep pool of the KT wetland. The deeper cells were chosen to ensure samplers were deployed within the water column to measure dissolved contaminants in the water. The PE strips (12.70 cm × 15.24 cm, 25 μm thickness, 0.5 g) were deployed in stainless steel mesh casings, and attached to an anchor-buoy system to allow for deployment at the centre of the deep cells, and to ensure the PE strips were submerged at approximately 0.3–0.5 m depths below the water surface.

Three deployments of passive samplers occurred, beginning on (1) 21 July 2017 (final deep pool) and 22 July 2017 (forebay) for a 37 day and 36 day deployment, respectively, (2) 28 August 2017 (forebay and final deep pool) for a 31 day deployment, and (3) 25 August 2018 (forebay) and 8 September 2018 (final deep pool) for 14 day deployments each. Since water was fully recycled through the KT wetland in 2018, the deployments in the forebay and final deep pool were done consecutively, i.e., samplers in the forebay were deployed from days 0–14, and samplers in the final deep pool were deployed from days 14–28.

The PE samplers were prepared and analysed by SGS Axys Analytical Services Ltd. (Sidney, BC, Canada). Samplers were stored and shipped in aluminium foil, sealed in a plastic bag, and shipped in a cooler with ice packs to maintain a temperature at <4 °C. The method of analysis of PE sampler devices follows USEPA Methods 1625B and 8270C/D. Instrumental analysis was performed by low-resolution mass spectrometry (LRMS) using an RTX-5 capillary GC column, which operates at a unit mass resolution in the electron impacts (EI) ionisation mode using multiple ion detection (MID), acquiring at least one characteristic ion for each target analyte and surrogate standard. Quantification of target analytes was performed using the isotope dilution method, and calculations were carried out using HP EnviroQuant and Prolab MS-Extended for targeting and quantification. Sample detection limits are available in the Supplementary Information (Tables S6–S8).

The reported concentrations provided by the laboratory were issued as units of mass of chemical per gram of polyethylene passive sampler (i.e., $C_{PE}$, ng/g). The partitioning behaviour of each chemical between the OSPW and the PE sampler was estimated using the calibration equation for PAHs reported by Lohmann [23]:

$$\text{Log } K_{PE\text{-}W} = 1.22 \, (\pm 0.046 \text{ SE}) \cdot \log K_{OW} - 1.22 \, (\pm 0.24 \text{ SE}) \quad (R^2 = 0.92; \text{ SE } 0.27) \tag{1}$$

This relationship correlates the polyethylene–water partitioning coefficient ($K_{PE\text{-}W}$), with the octanol–water partition coefficient ($K_{OW}$) for each test chemical. The log $K_{OW}$ for each PAH was obtained from EPISuite v4.11 [24].

Two field blanks per deployment consisting of clean polyethylene sheets were exposed to ambient air during sampler deployment and collection. Field blanks were wrapped in aluminium foil, sealed in a plastic freezer bag, and refrigerated at <4 °C between use. The average concentration of PAHs in the field blanks ($C_{F.Blank,i}$) were subtracted from the concentrations of PAHs measured in the deployed PE samplers ($C_{PE,i}$) to account for background exposure, i.e., $C^*_{PE,i} = C_{PE,i} - C_{F.Blank,i}$ [25]. Concentrations were assumed to be negligible (i.e., $C^*_{PE,i} = 0$) if the mean concentration of the analyte found in the field blanks exceeded the concentration of that analyte measured in the deployed passive samplers ($C_{F.Blank,i} > C_{PE,i}$).

Two performance reference compounds (PRCs) were impregnated into the polyethylene passive samplers during lab preparation: fluoranthene-d$^{10}$, and dibenzo(a,h)anthracene-d$^{14}$. These deuterated PRCs were used to evaluate the state of equilibrium between the water and PE by comparing day zero concentrations ($C_{PRC,0}$) to final concentrations ($C_{PRC,t}$) of the PRCs in the samplers after the deployment period (t, days). The mass transfer coefficients ($k_e$, d$^{-1}$) of the two PRCs were determined using:

$$k_e = \ln\left(\frac{C_{PRC,0}}{C_{PRC,t}}\right) \times \frac{1}{t} \tag{2}$$

To relate the depletion rate constant of the performance reference chemicals to the time to reach 95% equilibrium (i.e., $3/k_e$) between the water and the passive sampler for the target analytes, a linear relationship was developed between log $k_e$ and log $K_{OW}$ of the performance reference chemicals. This relationship was then used to determine $k_{e,i}$ from the log $K_{OW}$ of each target chemical i [26].

This $k_{e,i}$ was then used to account for the lack of achieving equilibrium within the sampling duration by calculating the dissolved concentration ($C_{WD}$) of each target chemical i in water as:

$$C_{WD,i} = \frac{C_{PE,i}^{*}}{K_{PE-W,i} \times \left(1 - e^{-k_{e,i}t}\right)}$$

(3)

Standard errors (SE) of $C_{WD}$ were derived through error propagation as:

$$SE_{C_{WD}} = \sqrt{\left(SE_{C_{PE,i}^{*}} \cdot \frac{\delta C_{WD}}{\delta C_{PE}^{*}}\right)^{2} + \left(SE_{K_{PE-W}} \cdot \frac{\delta C_{WD}}{\delta K_{PE-W}}\right)^{2} + \left(SE_{k_e} \cdot \frac{\delta C_{WD}}{\delta k_e}\right)^{2}}$$

(4)

where the SE of $C^{*}_{PE,i}$ was determined for each chemical from the measured concentrations of the target analytes in multiple passive samplers; the SE of $K_{PE-W}$ was estimated by applying the delta method to the log-transformed linear regression equation (Equation (1)); the SE of $k_{e,i}$ was determined from measurements of PRC concentrations in the PE samplers.

### 2.4. Data Analysis

Analytes were included in the data analysis if concentrations in the passive samplers exceeded the method detection limit (DL) in at least two of three replicates. Concentrations of PAHs in water below the DL were assigned a concentration equal to one-half of the chemical's DL (i.e., $C_i = DL/2$). This was applied to all final deep pool concentration measurements for 7-methylbenzo[a]pyrene in deployment one, one forebay concentration measurement for 2,6-dimethylnaphthalene in deployment two, and two final deep pool concentration measurements for acenaphthene in deployment three. The great majority of measured concentrations exceeded the DL. For these three substances, a range of average concentrations is provided to reflect the lower estimate (i.e., assuming concentration is zero) and upper estimate (i.e., assuming concentration is equal to the DL). A two-sample t-test assuming unequal variances was performed in JMP®, Version 13.1.0 [27] to detect statistical differences between mean dissolved aqueous concentrations measured in the forebay and final deep pool ($\alpha = 0.05$). Unequal variances in concentration measurements in the forebay and final deep pool were detected with a two-sided F-test ($p < 0.001$).

### 2.5. Wetland Treatment Performance Evaluation

#### 2.5.1. Concentration-Reduction

Changes to the concentration of test chemicals in the water ($E_{C,i}$) were derived from the concentration of each test chemical (*i*) in the passive samplers deployed in the influent wastewater (forebay; $C_{PE,i}^{eq.in}$) and treated effluent (final deep pool; $C_{PE,i}^{eq.out}$) as:

$$E_{C,i} = \left(1 - \frac{C_{PE,i}^{eq.out}}{C_{PE,i}^{eq.in}}\right) * 100$$

(5)

$E_{C,i}$ was estimated directly from the measured concentration of the target analyte i in the passive sampler ($C_{PE,i}$) to reduce error associated with converting concentrations in passive samplers to those in water. To correct for the different deployment durations of the samplers in forebay and final deep pool, equilibrium concentrations in the PE samplers were estimated using measured $k_{e,i}$ values with:

$$C_{PE,i}^{eq.} = \frac{C_{PE,i}^{*}}{\left(1 - e^{-k_{e,i}t}\right)}$$

(6)

### 2.5.2. Mass Removal

Mass removal of PAHs from the wetland was expressed in terms of a mass-loading removal efficiency ($E_{L,i}$) for each chemical i. $E_{L,i}$ was determined from the concentrations of freely dissolved analyte i in the influent ($C_{WD,i}^{*in}$) and effluent ($C_{WD,i}^{*out}$) water, and the corresponding volumetric flow rates (L/day) of water entering the forebay ($Q_{in}$) and leaving from the final deep pool ($Q_{out}$) of the KT wetland:

$$E_{L,i} = \left(1 - \frac{Q_{out} \cdot C_{WD,\,i}^{*out}}{Q_{in} \cdot C_{WD,\,i}^{*in}}\right) * 100 \tag{7}$$

$Q_{in}$ in the KT wetland was controlled and maintained at 432,000 L/day (5 L/s) for the duration of the study. $Q_{out}$ was estimated from the water budget: $Q_{out} = Q_{in} + Q_P - Q_{ET}$, where $Q_P$ is the precipitation rate (L/day) and $Q_{ET}$ is the evapotranspiration rate (L/day) in the KT wetland. The mass removal efficiency $E_{L,i}$ expresses the removal of dissolved contaminant mass from OSPW. $E_{L,i}$ differs from $E_{C,i}$ in that it accounts for the effects of changes in the volume of water in the wetland due to precipitation and evapotranspiration on concentration of the target chemical i in water. However, it should be stressed that because passive samplers measure only the concentration of dissolved contaminants in the influent and effluent, $E_{L,i}$ may not account for all mass of PAHs removed from the wetland, which includes both dissolved and undissolved (sorbed) PAHs.

Precipitation, temperature, and relative humidity data were obtained from historical records of the Fort McMurray, AB, Canada weather station available from Alberta Climate Information Services [28]. Total precipitation (P) was 56.0 mm, 38.9 mm, and 50.3 mm during each of the three deployment periods, respectively. Temperature and relative humidity were used to estimate daily evapotranspiration rates from the KT wetland using the Penman–Monteith equation. Evapotranspiration (ET) at the KT wetland was estimated to be 197 mm, 91.4 mm, and 52.9 mm during each of the three deployment periods, respectively. The volumetric rate of precipitation ($Q_P$) and evapotranspiration ($Q_{ET}$) were calculated using the total catchment area and surface area of wetland cells, respectively (i.e., $Q_P = SA_{catchment} \cdot P$; $Q_{ET} = \Sigma(SA_{cells}) \cdot ET$). The total catchment area ($SA_{catchment} = 15,264$ m$^2$) included everything within the external berms of the KT wetland. All precipitation within this area was assumed to enter the wetland as runoff. The total surface area of all wetland cells ($\Sigma(SA_{cells}) = 7894.6$ m$^2$) was estimated at the operating water levels, i.e., 1.7 m for deep pools and 0.4 m for shallow cells.

### 2.5.3. Toxicity

The change in OSPW toxicity was estimated using the chemical activity approach [29–31]. Chemical activity (*a*; unitless) is a thermodynamic quantity related to fugacity and chemical potential which for dilute solutions can be expressed as the ratio of the chemical's concentration (C; e.g., mol/m$^3$) to the chemical's solubility in the same media (S; e.g., mol/m$^3$), i.e., *a* = C/S. The application of chemical activity to assess toxicity for neutral hydrophobic organic chemicals has merit for two main reasons. First, when equilibrated, concentrations of chemicals in different media (e.g., the passive sampler, water and organisms in the water) exhibit similar chemical activities. Hence, when at equilibrium, contaminant concentrations in the passive samplers reveal the chemical activity of contaminants in the water and biota that reside in the wetland, or that may be exposed to the influent or effluent of the wetland. Unless the contaminants are biomagnified in the food-web, the concentration of the contaminants in the organisms exposed to the wetland water will at most approach the chemical activities in the water and the passive samplers. Biotransformation in the organisms can reduce the chemical activities of the parent compound below the chemical activity in the water. Secondly, studies have shown that a combined chemical activity of PAHs between 0.01 and 0.1 causes acute toxic effects through a mode of toxic action referred to as non-polar narcosis [32–34]. Hence, by converting concentrations of contaminants into chemical activities of contaminants, it is possible to assess whether acute toxic effects can be expected. This makes the chemical activity of PAHs in the water and passive samplers a useful metric for toxicity assessment of individual and mixtures of PAHs.

Chemical activity ($a_i$) of each of the detected PAHs in water entering and leaving the KT wetland was estimated from the concentration of PAHs in the polyethylene samplers ($C^*_{PE,i}$) and the solubility of each PAH in the polyethylene sheets ($S_{PE,i}$) as $a_i = C^{eq.}_{PE,i}/S_{PE,i}$. $S_{PE,i}$ was determined as $K_{PE-W,i} \times S_{water,i}$ where $S_{water,i}$ is the solubility of chemical i in water at 25 °C (reported in [20]). The summation of chemical activities for each individual PAH produces a total chemical activity of all analytes present in the OSPW influent ($\Sigma\, a^{in}_{PE,i}$) and effluent ($\Sigma\, a^{out}_{PE,i}$) of the KT wetland. By comparing the total chemical activities of PAH mixture in the water entering and leaving the KT wetland to the chemical activity threshold value for baseline toxicity ($a_0 = 0.01$), it is possible to assess whether the influent or effluent has the potential to be toxic to aquatic biota, and to what degree toxicity or toxicological risk has been reduced through wetland treatment. This approach can also be used for substances that exhibit a greater toxicity (or "excess toxicity") than the baseline toxicity as long as the degree of excess toxicity is known from empirical toxicity studies or other methods. For such substances, $a_i$ should be compared to $0.01/\gamma$, where $\gamma$ is the excess toxicity defined as toxicity greater than baseline toxicity (i.e., $\gamma > 1$). For determining the chemical activities of the PAHs in the PE samplers, changes in temperature during the two deployments were ignored and it was assumed that the mean temperature is adequate for determining the chemical activity of the PAHs.

## 3. Results and Discussion

### 3.1. Water Quality

Table 1 shows that BOD$_5$, Dissolved Inorganic Carbon (DIC), Dissolved Oxygen (DO), Dissolved Organic Carbon (DOC), pH, Total Dissolved Solids (TDS), Total Suspended Solids (TSS), turbidity, and water temperature of OSPW in the wetland were below the upper limits of the water quality targets (WQTs) listed in [35].

**Table 1.** Biochemical Oxygen Demand (BOD$_5$), conductivity, Dissolved Inorganic Carbon (DIC), Dissolved Oxygen (DO), Dissolved Organic Carbon (DOC), pH, Total Dissolved Solids (TDS), Total Suspended Solids (TSS), turbidity, and temperature (T$_{water}$) of OSPW in the Kearl Treatment Wetland during each deployment period.

| Parameter | Units | Deployment 1 2017A 13-Jul Forebay | Deployment 1 2017A 13-Jul FDP | Deployment 2 2017B 17-Aug Forebay | Deployment 2 2017B 17-Aug FDP | Deployment 3 2018 26-Aug Forebay | Deployment 3 2018 19-Sep Forebay | WQT [+] |
|---|---|---|---|---|---|---|---|---|
| BOD$_5$ | mg/L | <2.0 | <2.0 | <2.0 | <2.0 | — | 2 | 2.4 |
| Conductivity | µS/cm (±0.5%) | 811 | 795 | 955 | 919 | 1700 | 1700 | 799 |
| DIC | mg/L | 74 | 75 | 74 | 75 | 46 | 52 | |
| DO | mg/L (±0.2) | 4.57 | 4.46 | 5.09 | 6.39 | 5.04 | 9.99 [d] | 1.44 [m] |
| DOC | mg/L | 18 | 18 | 18 | 18 | 20 | 20 | 63.1 |
| pH | pH (±0.2) | 7.92 | 7.77 | 7.97 | 7.49 | 7.40 | 8.24 | 6.0–10.8 |
| TDS | mg/L | 680 | 680 | 680 | 680 | 860 | 1200 | |
| TSS | mg/L | <1.0 | <1.0 | <1.0 | <1.0 | 12 | 1.6 | 82.2 |
| Turbidity | NTU (±2%) | 0.84 | 1.37 | 0.84 | 1.11 | 45 | 1.5 | 77 |
| T$_{water}$ | °C (SD) | 20.7 (2.1) | | 20.6 (2.1) | | 10.7 (2.2) | | 25.3 |

[+] Water Quality Target (peak target), [35]. <—below the reported detection limit. [m]—minimum value. [d]—data collected on 22-Sep-18. ——parameter not analysed. FDP—Final Deep Pool. (±)—Instrument accuracy.

DO in water remained constant throughout the wetland during deployment one (DO$_{forebay}$ = 4.57 mg/L; DO$_{FDP}$ = 4.46 mg/L) and increased upon passage through the wetland during deployments two (DO$_{forebay}$ = 5.09 mg/L; DO$_{FDP}$ = 6.39 mg/L) and three (DO$_{forebay}$ = 5.04 mg/L;

$DO_{FDP}$ = 9.99 mg/L). The DO concentrations indicate that reoxygenation of the water occurs as it passes over the internal berms between wetland cells.

The average recorded water temperature in the KT wetland was 20.7 (SD 2.1) during deployment one, 20.6 (SD 2.1) during deployment two, and 10.7 (SD 2.2) during deployment three. In a study using a surface flow mesocosm wetland, temperature changes from 18.4 °C to 11.3 °C showed no significant effects on overall heterotrophic activity [36], possibly due to the diversity in microbial communities contributing to contaminant removal at different temperatures [37]. However, the effect on water temperature on heterotrophic activity in wetlands remains an area requiring further investigation.

Warmer air temperatures were recorded during deployment one ($T_{avg.}$ = 17.9 °C; SD 2.7) compared to deployments two ($T_{avg.}$ = 12.2 °C; SD 4.5) and three ($T_{avg.}$ = 5.8 °C; SD 4.3). Modelling work using a modified Penman–Monteith equation demonstrated that even a 3 °C change in air temperature can induce a 14% change in potential evapotranspiration rates [38]. Therefore, differences in air temperatures among the deployments are expected to have significant effects on evapotranspiration of water from the wetland. Estimated rates of evapotranspiration range between 2.5 to 9.8, 0.6 to 5.8, 1.0 to 3.1 mm/day and total evapotranspiration was 53.7, 24.9, and 14.4 $m^3$/day in deployments one, two, and three, respectively (Table S4).

The conductivity of OSPW ranged from 795 to 811 μS/cm in deployment one, 919 to 955 μS/cm in deployment two and was 1700 μS/cm in deployment three and appeared to remain constant throughout the wetland in all three deployments. Water quality targets for conductivity (WQT = 799 μS/cm; [35]) were exceeded in all deployments. Elevated levels of conductivity are common in OSPW due to naturally high levels of sodium, bicarbonate, chloride, and sulphate found in waters around the Athabasca oil sands. Since conductivity showed no significant effects on three floating wetland species at values up to 3000 μS/cm [39], significant effects on treatment performance as a result of the measured conductivities are not expected in the KT wetland. However, at a conductivity of 4000 μS/cm, which is much greater than that in the KT wetland, a 44.4–67.9% reduction in $BOD_5$ in a continuous flow constructed wetland system with cattails (*Typha angustifolia*) and Asian crabgrass (*Digitaria bicornis*) was observed [40].

Total dissolved solids (TDS) ranged between 680 mg/L to 1200 mg/L and were lower than those typically observed in OSPW (i.e., 2000–2500 mg/L; [41]) and within the wide range of concentrations of dissolved solids in surface waters (i.e., 240 mg/L–279,000 mg/L) of the Athabasca oil sands region [42]. No WQT has been issued for TDS. Removal efficiencies of organic contaminants in treatment wetlands tend to decrease as salinity increases due to effects on plants, microorganisms, and substrates [43].

The pH of OSPW ranged between 7.77 and 7.92 in deployment one, 7.49 and 7.99 in deployment two, and 7.40 and 8.24 in deployment three. Alkaline pH levels ranging from 8.0–8.4 have been measured in OSPW from other facilities in the Athabasca oil sands region [41] and are common when alkaline reagents such as sodium hydroxide are added to the process water to improve hydrocarbon extraction efficiency.

TSS concentrations in the OSPW were below detection limits in deployments one and two in 2017. This was due to sedimentation in the overburden disposal pond before water was introduced into the KT wetland. During deployment three in 2018, water was introduced into the wetland during a single rapid pumping event (<24 h) at rates of 4.0–4.7 $m^3$/min, which caused turbulence and resuspension of sediment in the source water pond. Since no sedimentation pond was used as a preliminary settling basin, OSPW entering the KT wetland in 2018 contained higher concentrations of TSS than in the 2017 deployments. Within the wetland, TSS reduced by 87% after 24 days of recycling demonstrating the capacity for particulate settling in the KT wetland.

### 3.2. Polycyclic Aromatic Hydrocarbons in Influent

During deployments one and two, freely dissolved concentrations of PAHs in water were highest for chrysene (1.28 (SE 0.35) and 0.37 ng/L (SE 0.039)), fluoranthene (0.92 (SE 0.033) and 0.54 ng/L (SE 0.011)), phenanthrene (0.78 (SE 0.075) and 0.88 ng/L (SE 0.047)), and pyrene (5.41 (SE 0.12) and

3.52 ng/L (SE 0.066)), respectively (Figures 2 and 3). During deployment three freely dissolved PAH concentrations entering the wetland were highest for acenaphthene (2.88 (SE 0.082) ng/L), fluorene (1.34 (SE 0.027) ng/L), phenanthrene (3.83 (SE 0.085) ng/L), pyrene (2.24 (SE 0.13) ng/L), and retene (2.03 (SE 1.4) ng/L) (Figure 4).

With the exception of pyrene in the influent of deployment one, concentrations of PAHs in the influent were below the criteria for the protection of aquatic life in Canada (i.e., [44]) and the water quality targets for the Muskeg River (i.e., [35]). Concentrations of PAHs in the influent were similar to concentrations measured by Environment and Climate Change Canada (ECCC) in the Athabasca, Peace, and Slave rivers [45] (Figure S1). The concentration of pyrene in the influent exceeded water quality targets in the Muskeg River Interim Management Framework [35] and points to the need to develop remediation strategies to reduce contaminant concentrations to levels that meet environmental quality guidelines and protect wildlife in or visiting the wetland.

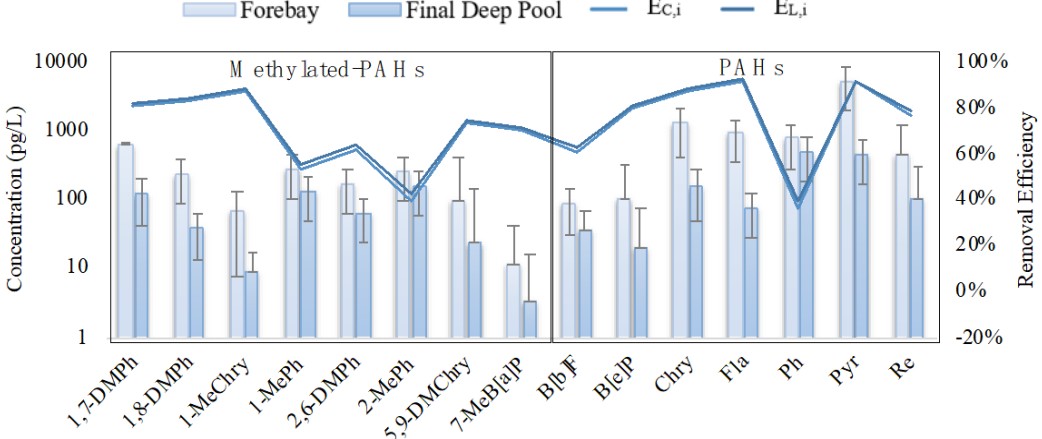

1,7-DMPh = 1,7-Dimethylphenanthrene; 1,8-DMPh = 1,8-Dimethylphenanthrene; 1-MeChry = 1-Methylchrysene; 1-MePh = 1-Methylphenanthrene; 2,6-DMPh = 2,6-Dimethylphenanthrene; 2-MePh = 2-Methylphenanthrene; 5,9-DMChry = 5,9-Dimethylchrysene; 7-MeB[a]P = 7-Methylbenzo[a]pyrene; B[b]F = Benzo[b]fluoranthene; B[e]P = Benzo[e]pyrene; Chry = Chrysene; Fla = Fluoranthene; Ph = Phenanthrene; Pyr = Pyrene; Re = Retene

**Figure 2.** Dissolved aqueous concentrations and removal efficiencies of PAHs in the Kearl Treatment Wetland during deployment one (2017A). Error bars represents standard errors of the mean.

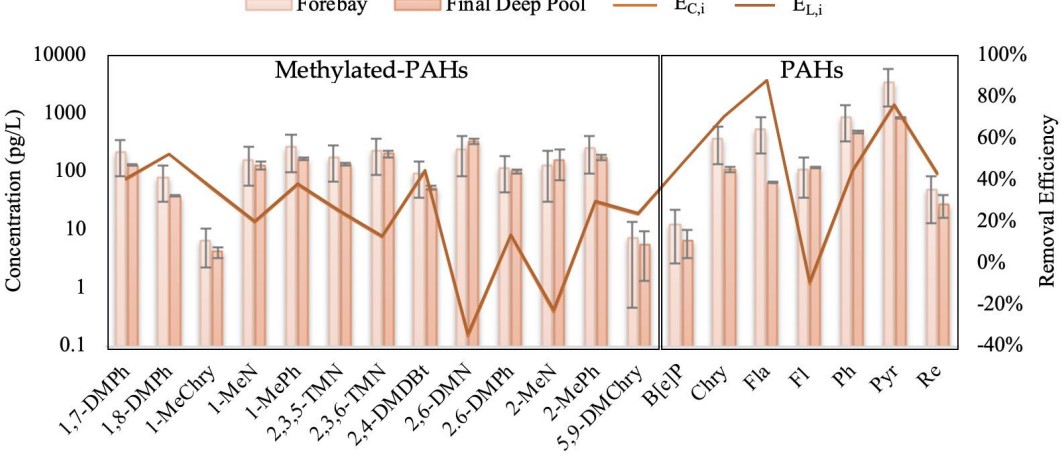

1,7-DMPh = 1,7-Dimethylphenanthrene; 1,8-DMPh = 1,8-Dimethylphenanthrene; 1-MeChry = 1-Methylchrysene; 1-MeN = 1-Methylnaphthalene; 1-MePh = 1-Methylphenanthrene; 2,3,5-TMN = 2,3,5-Trimethylnaphthalene; 2,3,6-TMN = 2,3,6-Trimethylnaphthalene; 2,4-DMDBt = 2,4-Dimethyldibenzothiophene; 2,6-DMN = 2,6-Dimethylnaphthalene; 2,6-DMPh = 2,6-Dimethylphenanthrene; 2-MeN = 2-Methylnaphthalene; 2-MePh = 2-Methylphenanthrene; 5,9-DMChry = 5,9-Dimethylchrysene; B[e]P = Benzo[e]pyrene; Chry = Chrysene; Fla = Fluoranthene; Fl = Fluorene; Ph = Phenanthrene; Pyr = Pyrene; Re = Retene

**Figure 3.** Dissolved aqueous concentrations and removal efficiencies of PAHs in the Kearl Treatment Wetland during deployment two (2017B). Error bars represents standard errors of the mean.

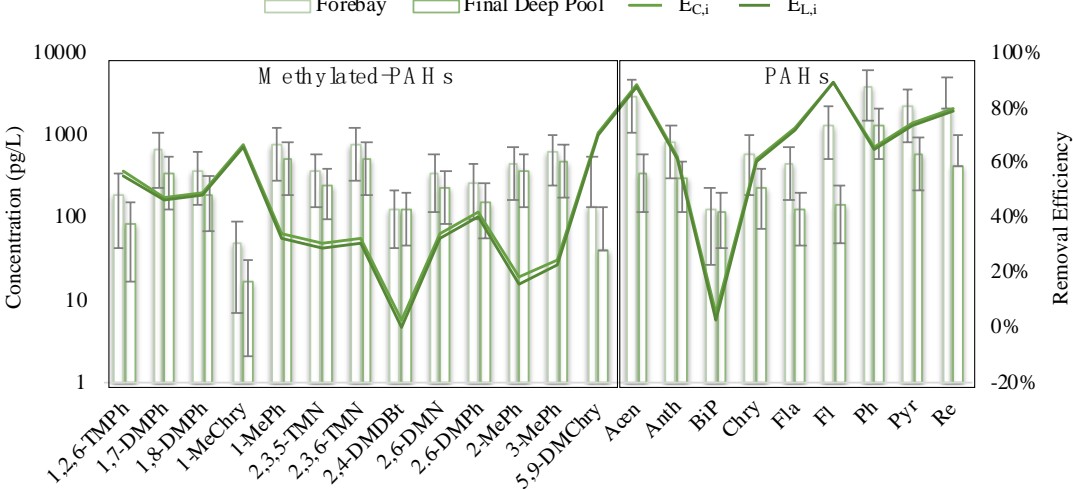

1,2,6-TMPh = 1,2,6-Trimethylphenanthrene; 1,7-DMPh = 1,7-Dimethylphenanthrene; 1,8-DMPh = 1,8-Dimethylphenanthrene; 1-MeChry = 1-Methylchrysene; 1-MePh = 1-Methylphenanthrene; 2,3,5-TMN = 2,3,5-Trimethylnaphthalene; 2,3,6-TMN = 2,3,6-Trimethylnaphthalene; 2,4-DMDBt = 2,4-Dimethyldibenzothiophene; 2,6-DMN = 2,6-Dimethylnaphthalene; 2,6-DMPh = 2,6-Dimethylphenanthrene; 2-MePh = 2-Methylphenanthrene; 3-MePh = 3-Methylphenanthrene; 5,9-DMChry = 5,9-Dimethylchrysene; Acen = Acenaphthene; Anth = Anthracene; BiP = Biphenyl; Chry = Chrysene; Fla = Fluoranthene; Fl = Fluorene; Ph = Phenanthrene; Pyr = Pyrene; Re = Retene

**Figure 4.** Dissolved aqueous concentrations and removal efficiencies of PAHs in the Kearl Treatment Wetland during deployment three (2018). Error bars represents standard errors of the mean.

### 3.3. Wetland Treatment Performance

Treatment in the KT wetland resulted in statistically significant reductions in dissolved aqueous concentration for 14 out of 15 PAHs during deployment one, 15 out of 20 PAHs in deployment two, and 19 out of 22 PAHs in deployment three (Figures 2–4). Freely dissolved concentrations of pyrene in OSPW reduced from 5.41 (SE 0.12) ng/L to 0.46 (SE 0.044) ng/L during deployment one, from 3.52 (SE 0.066) ng/L to 0.84 (SE 0.012) ng/L during deployment two, and from 2.24 (SE 0.13) ng/L to 0.58 (SE 0.0087) ng/L during deployment three. This corresponds to an $E_{C,i}$ for pyrene of 91, 76, and 75% and an $E_{L,i}$ for pyrene of 92, 76, and 74% for deployments 1–3, respectively (Figures 2–4). Values of $E_{C,i}$ and $E_{L,i}$ are similar because water inflows and outflows in the wetland were well balanced. Pyrene measured in deployment one reduced to concentrations that did not exceed the water quality guideline for pyrene [35].

Both $E_{C,i}$ and $E_{L,i}$ for individual PAHs varied substantially, i.e., from 0% (no statistical differences in concentration through the wetland) for certain methylated PAHs to 92% (for $E_{C,i}$) and 93% (for $E_{L,i}$) for fluoranthene during the first deployment. The mean $E_{C,i}$ for all analytes was measured at 72 (SE 4.7)%, 32 (SE 6.9)%, and 50 (SE 5.4)% for each of the three deployments, respectively and closely matched the mean $E_{L,i}$ of 73 (SE 4.4)% for deployment one, 32 (SE 6.9)% for deployment two, and 49 (SE 5.5)% for deployment three. The overall reduction in concentration of all PAHs combined ($\Sigma C_{PAH}$) ranged from 56% (deployment two) to 82% (deployment one). The reduction in total mass of all measured PAHs was 83 (SE 37)%, 54 (SE 16)%, and 64 (SE 19)%, in deployments one, two, and three, respectively. The reductions in mass and concentrations of the PAHs, and hence $E_{L,i}$ and $E_{C,i}$ in each deployment were similar because the net change of water in the wetland was close to zero with only small gains and losses of water in the KT wetland. These reductions in concentrations align with those found in other studies. For example, [46] measured an $E_C$ for the combined sum of 16 PAHs of 56% after 24 h in a pilot-scale surface flow wetland that treated highway runoff in southern Greece. In a pilot-scale vertical flow constructed wetland system in Munich, Germany [47], a 99% reduction in the concentration of phenanthrene in artificial wastewater (8 µg/L) was achieved over 14 days. The present study measured a reduction in the concentration of phenanthrene in OSPW influent (i.e., 0.77 ng/L and 3.83 ng/L) in the KT wetland of 36 to 66% over 14 days.

No statistically significant differences between the influent and effluent concentrations were observed for nine PAHs (i.e., benzo[b]fluoranthene (deployment one), 1-methylnaphthalene,

2-methylnaphthalene, 2,3,6-trimethylnaphthalene, 2,6-dimethylnaphthalene, fluorene (deployment two), and 2,4-dimethyldibenzothiophene, 2,6-dimethylnaphthalene, and biphenyl (deployment three). Five of these chemicals are naphthalene-based PAHs with low molecular weights, high aqueous solubilities, and relatively low log $K_{OW}$ (3.86–4.73) when compared to the other test chemicals in this study. High water solubility and low log $K_{OW}$ make substances less susceptible to mechanisms of chemical removal in rooting media, biofilm, and vegetation because a higher proportion of the chemical remains in the water phase [48].

Pyrene, fluoranthene, and chrysene consistently exhibited the highest $E_{L,i}$ in all three deployments whereas 2-methylphenanthrene and 2,6-dimethylphenanthrene consistently showed the lowest $E_{L,i}$ among the three deployments (Figure 5). This consistent pattern of relative treatment efficiency of these PAHs further indicates that chemical make-up and properties play an important role in wetland treatment. Furthermore, since the operation of the wetland was replicated in all three deployments, the considerable variation in $E_{L,i}$ of individual PAHs between deployments shows that environmental conditions and water quality characteristics also play an important role in wetland treatment.

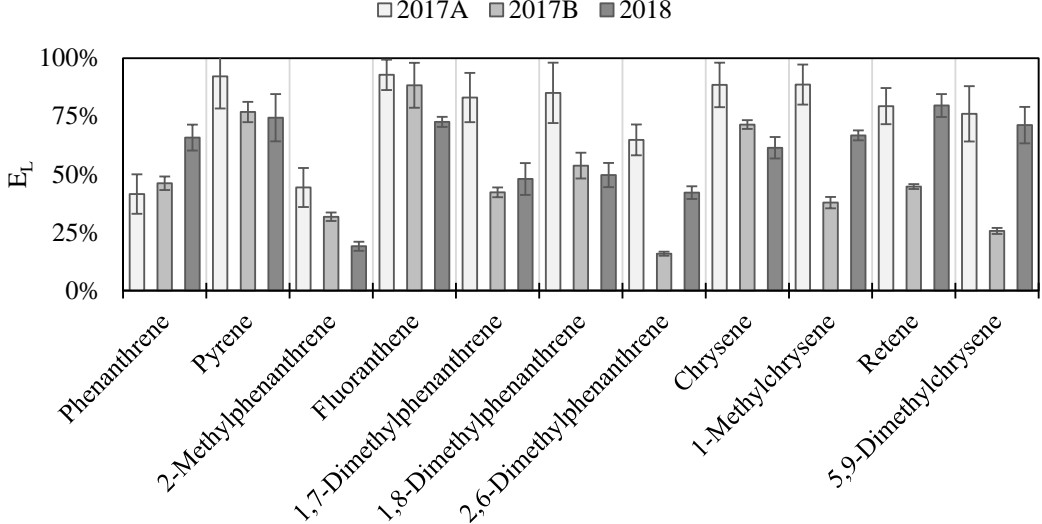

**Figure 5.** Mass-removal efficiency of PAHs measured in all three deployments of PES in the Kearl treatment wetland.

Toxicity

Chemical activities of individual PAHs (Table S5) were all far below the baseline toxicity value ($a_0$) of 0.01. This indicates that individual PAHs in both the influent and the effluent of the wetland were below concentrations that cause acute toxic effects in biota. Retene, chrysene, and pyrene exhibited the highest chemical activities in the KT wetland throughout the study. The highest chemical activities for all three of these chemicals were found in the forebay during deployment one, where $a_{retene} = 5.2 \times 10^{-5}$ (SE $1.7 \times 10^{-6}$), $a_{chrysene} = 4.8 \times 10^{-5}$ (SE $5.1 \times 10^{-7}$), and $a_{pyrene} = 4.0 \times 10^{-5}$ (SE $8.6 \times 10^{-7}$). These chemical activities reduced to $1.2 \times 10^{-5}$ (SE $3.1 \times 10^{-5}$) for retene, $6.1 \times 10^{-6}$ (SE $1.6 \times 10^{-5}$) for chrysene, and $3.4 \times 10^{-6}$ (SE $4.0 \times 10^{-5}$) for pyrene in the final deep pool of the wetland, corresponding to reductions in chemical activity of 77, 87, and 91%, respectively. The largest reduction in chemical activity was observed for fluoranthene during deployment one, with a reduction of 92%.

The total chemical activity ($\Sigma a$) for all analytes included in the analysis was also below the baseline toxicity value ($a_0$) of 0.01 suggesting concentrations of PAHs in water are below the threshold for acute effects in aquatic biota due to non-polar narcosis. These results are in agreement with the toxicity experiments with the influent OSPW performed by Maxxam Analytics (unpublished data), which showed no acute toxicity of the OSPW to rainbow trout. $\Sigma a$ was highest during deployment three

($n_3$ = 22 PAHs) at $3.8 \times 10^{-4}$ (SE $3.4 \times 10^{-6}$) and reduced to $1.0 \times 10^{-4}$ (SE $6.4 \times 10^{-5}$) in the final deep pool (Figure 6). Despite more PAHs being detected during deployment two (i.e., 20 PAHs) than during deployment one (i.e., 15 PAHs), $\Sigma a$ was higher during deployment one, largely because average concentrations of PAHs were greater in deployment one than in deployment two. Since the combined chemical activities of the PAHs investigated in this study were far below 0.01, both the influent and effluent are not expected to cause PAH related acute toxicity. Hence, a toxicity reduction as a result of wetland treatment cannot be determined. However, if influents with higher concentrations of PAHs are used, treatment may cause a toxicity reduction that can be monitored using a passive sampling approach.

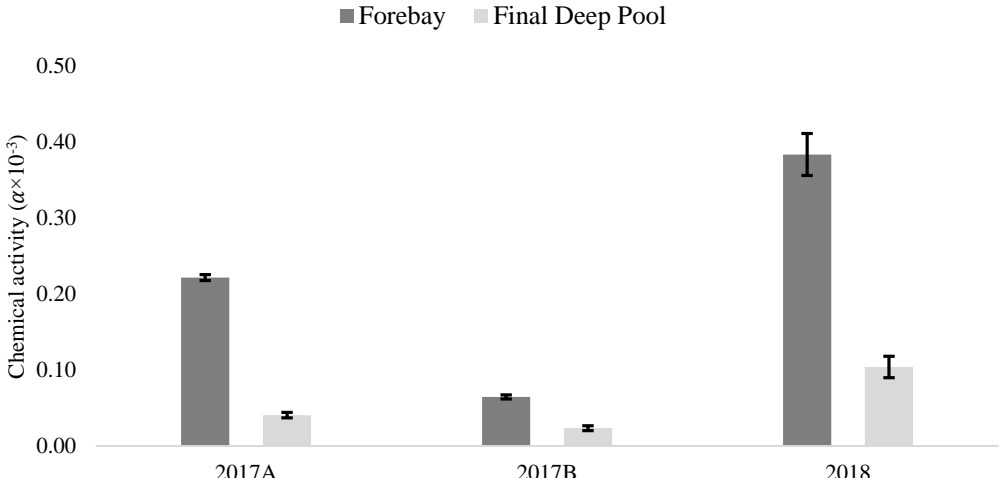

**Figure 6.** Total chemical activity of ΣPAHs in the polyethylene samplers in the forebay and final deep pool during each deployment.

Some limitations of this approach are noteworthy. First, the total chemical activity estimate is limited to the contaminants that were analysed in this study. Other contaminants not quantified in this study are likely to add to the total chemical activity. These contaminants include ionisable and hydrophilic substances such as naphthenic acids which have been recognized as the primary source for aquatic toxicity in OSPW [49]. This means that the total chemical activity of bulk OSPW is likely underestimated in this study. Redman et al. [50] addressed this problem by analysing both neutral organics such as PAHs, and ionisable substances such as naphthenic acids using a non-specific passive sampling technique where the complex mixture of contaminants in OSPW is analysed as a single contaminant using gas-chromatography with flame-ionisation detection. Second, the passive sampling approach applies an equilibrium assumption to equate the chemical activity in the water to that in the organism. This assumption ignores potential biotransformation of PAHs that may occur within wetland biota and therefore chemical activity of PAHs and potential for toxicity in OSPW is likely overestimated. Despite these limitations of the passive sampling approach, it is encouraging that the results are consistent with the standard toxicity tests. Further testing of the passive sampling approach outlined in Redman et al. [50] for toxicity testing may be useful in evaluating the effect of wetland treatment on toxicity in wildlife without undue reliance on animal testing.

## 4. Conclusions

This study shows that a surface treatment wetland is able to substantially reduce concentrations of PAHs in OSPW in Alberta, Canada. Similar observations were observed in a subsurface flow treatment wetland treating municipal effluents in Singapore [48], and in a free water surface flow wetland treating highway runoff in southern Greece [46]. The combined information suggests that wetland treatment under aerobic conditions is a suitable method for wetland treatment of PAHs in different types of wastewaters under different climatic conditions.

However, large differences in concentration-reduction were observed among individual PAHs. While chrysene and fluoranthene were readily removed from OSPW, concentrations of certain methylated PAHs did not show statistically significant declines in concentration as a result of wetland treatment. This means that wetland treatment does not universally apply to all PAHs or to organic pollutants in general. Further work is needed to investigate the role of chemical structure on wetland treatment and to develop structure-activity relationships that can be used to best match wetland treatment potential to pollutant types.

Since treatment efficiencies at 10 °C in deployment 3 were greater than those at 20 °C in deployment 2, there does not appear to be a simple relationship where an increase in temperature corresponds to an increase in treatment efficiency. Treatment efficiency appears to be controlled by a combination of chemical, environmental and wetland design characteristics. Models like Cancelli et al. [48] can help to better understand and anticipate the response of wetland treatment to these factors. The findings of our study do suggest that wetland treatment is feasible in both cold and warm climates. Whether wetland treatment is feasible in conditions below 0 °C remains to be investigated, but wetland design may be adapted to provide treatment capacity during very cold parts of the year.

Our study also demonstrated that wetland treatment can in some cases be an effective method for wastewaters to achieve water quality objectives. The latter is an important incentive for the development of treatment wetlands. In our study, the concentration of pyrene in one deployment reduced to levels below the Muskeg River Interim Management Framework [35] water quality target as a result of wetland treatment.

Our study also indicates that passive sampling is a feasible (and relatively cost effective) method for monitoring wetland treatment capacity of organic pollutants over an extended period of time. The monitoring results provide time integrated information on treatment efficiency that is useful for tracking not only treatment efficiency but also the toxicity of the treatment wetland environment to aquatic organisms that may reside or visit the wetland.

Overall, wetland treatment has shown to be able to reduce PAH concentrations, suggesting this treatment technology may provide a feasible option for OSPW treatment. To better understand treatment efficiency for these systems, we aim to use the information of this study as well as a similar study on naphthenic acids to develop and test a model of the treatment efficiency of wetlands that can account for the effect of chemical properties, wetland design characteristics and environmental conditions.

**Supplementary Materials:** The following are available online at http://www.mdpi.com/2076-3298/7/9/64/s1. Table S1. Dissolved PAH concentrations at the forebay and final deep pool for deployment one (21 July–29 August 2017). Table S2. Dissolved PAH concentrations at the forebay and final deep pool for deployment two (29 August–29 September 2017). Table S3. Dissolved PAH concentrations at the forebay and final deep pool for deployment three (25 August–22 September 2018). Table S4. Inflow rate ($Q_{in}$), precipitation rate ($Q_P$), evapotranspiration ($Q_{ET}$), and outflow rate ($Q_{out}$) of water at the Kearl Treatment Wetland during each deployment period. Table S5. Chemical activity ($\alpha$) of PAHs in polyethylene passive samplers. Table S6. PAH concentrations in OSPW collected from the forebay and final deep pool of the Kearl Treatment Wetland for deployment one. Table S7. PAH concentrations in OSPW collected from the forebay and final deep pool of the Kearl Treatment Wetland for deployment two. Table S8. PAH concentrations in OSPW collected from the forebay and final deep pool of the Kearl Treatment Wetland for deployment three. Figure S1. Comparison of concentrations of select PAHs in Oil Sands Process-Affected Water entering the Kearl Treatment Wetland during deployments one, two, and three to Environment and climate Change Canada (2015) environmental measurements and Alberta Environment (2008) WQTs.

**Author Contributions:** Conceptualization, F.A.P.C.G.; Data curation, A.M.C.; Funding acquisition, F.A.P.C.G.; Investigation, A.M.C.; Methodology, A.M.C. and F.A.P.C.G.; Project administration, F.A.P.C.G.; Supervision, F.A.P.C.G.; Writing – original draft, A.M.C. and F.A.P.C.G.; Writing – review & editing, A.M.C. and F.A.P.C.G. All authors have read and agreed to the published version of the manuscript.

**Funding:** This work was funded by the National Sciences and Engineering Research Council of Canada—Collaborative Research and Development grant with Imperial Oil Resources Limited (NSERC CRD 525587-18), and through a Mitacs© Accelerate Internship with Imperial Oil Resources Limited (IT08005 and IT09875).

**Acknowledgments:** We thank Asfaw Bekele, Michelle Young, and operations personnel at Imperial Oil Resources Limited for their technical support and guidance in planning and execution.

**Conflicts of Interest:** The authors declare no conflict of interest.

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
