# Peer review of "Treatment of Polycyclic Aromatic Hydrocarbons in Oil Sands Process-Affected Water with a Surface Flow Treatment Wetland"

_environments, doi:10.3390/environments7090064_

Round 1

Reviewer 1 Report

The paper has a lot of good information in it and it's well written. My main concern is that there were only 3-4 sample dates which is not sufficient to characterize the effectiveness of a treatment wetland at removing pollutants. The authors should acknowledge this shortcoming and state that more extensive data collection is needed to confirm the preliminary results of this study. In terms of overall paper structure, the intro doesn't sufficiently place this study within the context of other treatment wetland studies. The conclusions section is also very brief. 

More specific comments: 

Intro, Lines 27-30: Many people will not be familiar with oil sand processes so you should explain that briefly in 1-2 sentences.

Lines 93-95: only three dates of sampling are insufficient to characterize treatment efficiency. The results could be very different in the spring or fall, for example. the low sample size is not sufficient to characterize the variability of removal

Lines 340-345 Figures 2,3,4.  Is it possible to group the PAHs into similar groups? it would make the graphs easier to read and interpret. 

Conclusion, Lines 438-443. in extending your results to other settings it would be good to discuss how applicable this study is to other types of wetlands, in other regions or climates, that receive PAHs in runoff? This section if very thin.

Reviewer 2 Report

Comments to authors

The research results presented by the authors in “Treatment of polycyclic aromatic hydrocarbons in oil sands process-affected water with a surface flow treatment wetland” are interesting, however not clear enough. Please think about how to present your results to make them more attractive and understandable. I  have several detailed comments:  

Abstract

The aim of the research should be clearly stated.

Line 14-15 “Treatment efficiencies of individual PAHs ranged between essentially 0% for some PAHs to 95% for others”- please specify what kind of PAHs

Line 17: please explain abbreviation OSPW

Line 18 “local regulatory water quality criteria”- not clear enough

Introduction

Line 46 – add abbreviation BOD etc. and use them in the following tekst.

Should be BOD5 instead of BOD (should be corrected in the whole paper)

Materials and Methods

Line 59 – just use KT, don’t explain it again (it was done a few lines above).

Please extend this chapter – please give some more information about Oil Sand Process affected Waters (OSPW)

Line 71 suspended sediments or suspended solids ?

Is it possible to present also photo of the research installation during study period?

Lines 93-105 – just use abbreviations of BOD5, do not repeat it explanations.

Water quality monitoring Please explain why you did not measure COD.

Results

Lines 253 – just use abbreviations of BOD5, do not repeat it explanations.

Table 1 Parameter instead Metric

BOD5 not BOD

Improve figure 2 (especially description). Just look at figure 3 and 4.

Conclusions

Chapter conclusion is definitely too short for high quality research paper. Please add strong detailed conclusions derived from your research.

Reviewer 3 Report

The authors examined the treatment efficiency of wetlands in the removal of PAHs in OSPW. The manuscript in general is of interest to the readership of Environments. 

The authors need justify the focus of PAH in this study, and to address the relative effects of different factors in affecting the PAH levels, e.g. environmental and climatic conditions, wastewater characteristics, in the introduction and discussion. 

Introduction - pls justify the study of PAH, any previous studies of PAHs in OSPW?

line 167-168 please justify "Concentrations of PAHs in water below the MDL were assigned a concentration equal to one-half of the chemical’s DL (i.e. Ci = DL/2)." what were the grounds? 

line 307- Table 1 pls add SD of the parameters? 

Line 381-383 pls add supporting evidence to your statement of "High water solubility and low log KOW make substances less susceptible to mechanisms of chemical removal in rooting media, biofilm, and vegetation because a higher proportion of the chemical remains in the water phase."

line 388-389 With reference to the influence of the varying environmental conditions on the levels of PAHs, pls justify the validation of the results of the study. 

The considerable variation in EL,i of
388 individual PAHs (Fig. 5) shows that environmental conditions also play an important role in wetland
389 treatment.

Round 2

Reviewer 3 Report

The responses are generally acceptable, there seems to be a gap between their responses to Comment 3 and 5 though. Some addresses/explanations are warranted to illustrate the possible role of environmental conditions in the wetland treatment in this study.

When as stated in the response to Comment 3 -  "there was insignificant variation in environmental conditions to justify multiple measurements. True replicate measurements of the water quality parameters were not collected", in the response to Comment 5, however, it states that "environmental conditions .... also play an important role in wetland treatment.“
